# Performance Analysis of Compound Rubber and Steel Slag Filler Modified Asphalt Composite

**DOI:** 10.3390/ma12162588

**Published:** 2019-08-14

**Authors:** Yongjie Xue, Hui Zhao, Xintong Wei, Yunya Niu

**Affiliations:** State Key Laboratory of Silicate Materials for Architectures, Wuhan University of Technology, Wuhan 430070, China

**Keywords:** modified asphalt, crumb rubber, steel slag filler, rheological properties, CAM model, burgers model

## Abstract

A new treatment method of combined crumb rubber and steel slag modifier for asphalt binders was proposed in this work. The viscosity, rheological properties, and thermogravimetric analysis of modified asphalt mortar were then investigated. The modified asphalt composite was prepared in laboratory by two steps. Rubber powder was firstly added into hot asphalt flux to make rubber modified asphalt (RA), and then RA binders were wrapped with steel slag powder by granulation machine to make compound rubber and steel slag filler modified asphalt composite (RSAC). Test results showed that the viscosity–temperature susceptibility of RSAC was superior to that of modified asphalt binder with only one additive. The softening point differences of RSAC was 2.1 °C. The complex modulus and phase angle were significantly influenced by the addition of steel slag fillers. Creep tests show that a better anti-permanent deformation performance of RSAC can be obtained, which means a better low temperature performance could be predictable. The CAM (Christensen-Anderson-Marasteanu) and Burgers models can be used to describe the change of complex modulus and viscous-elasticity performance of RSCA. The lower value of *m_e_* (0.6344) and *R* (0.1862) from the CAM model indicated that RSAC was slightly related to the sensibility of frequency. The higher value of *λ_∞_* and *E*_0_ of RSAC indicated a better ability of shear-creep resistance.

## 1. Introduction

Road construction is a major part of infrastructure. During road construction, asphalt pavements are widely used on highways due to advantages such as the driving safety and comfort offered by asphalt mixtures [1]. Several kinds of raw materials are usually used in asphalt pavement, such as asphalt binders, aggregates, and mineral fillers [2]. The combination of asphalt binder, mineral filler, and fine aggregate consists of asphalt mortar, which mainly determines the final mechanical and durable properties of asphalt pavement. A previous study has reported that the higher temperature anti-rutting or low-temperature anti-cracking properties of asphalt mixtures are highly related to asphalt mortar [3]. Hence, lots of scientific research has focused on the modifier of asphalt binders and mineral filler of asphalt mortar for improving the properties of asphalt mixture. 

Recently, crumb rubber from scrap tires has been wildly used in the field of asphalt modification. The reutilization of the end-of-life tires to modify virgin binder provides a promising way to significantly reduce the number of land-filled tires [4]. Besides, it is known that rubber can improve the performances of asphalt binders by optimizing the property of asphalt through the swelling, degradation, and a series of complex physico-chemical processes [5,6]. According to previous findings, when adding crumb rubber, the low-temperature performance, skid resistance, and fatigue properties of modified asphalt have been improved. Additionally, adding crumb rubber significantly improved the rutting resistance, temperature susceptibility, and moisture susceptibility of asphalt mixture [7,8], as well as even long-term pavement service performance characteristics, such as driving safety and comfort [9]. Therefore, the waste rubber has been added into asphalt as a modifier to prepare rubber modified asphalt (RA) worldwide.

To further improving the performance of asphalt mixtures, mineral fillers such as fly-ash, cement, silica fume, bag-house fines, and hydrated lime have been used in asphalt mixtures for the promotion of rutting resistance and moisture damage resistance [10]. It has been found that the permanent deformation of asphalt mixtures can be improved by the type and amount of filler [11,12,13]. Steel slag powder can be used as a mineral filler in asphalt mixture [14]. Lots of research has focused on the reutilization of steel slag as a coarse aggregate to replace the natural mineral aggregates in design and preparation of asphalt mixture due to its high abrasion resistance and skid resistance [15,16]. Additionally, excessively reducing the consumption of steel slag depends not only on reutilization as coarse aggregate, but also as a mineral filler in asphalt mixture. Consequently, it can minimize the overall environmental impact of steel slag. However, the feasibility of application of steel slag are still unknown. It is crucial to the design of long-life asphalt pavement to understand the performance and optimal additional amount of filler, especially for a novel mineral filler prepared using solid waste materials.

Generally, for any process of modification of asphalt binders by crumb rubber or mortar by mineral fillers, the rubber or fillers are directly added with asphalt binder before the preparation of asphalt mixtures. Few examples of research have focused on the development of compound crumb rubber mineral filler asphalt composite prior to prepare of asphalt mixtures [17,18,19,20,21]. Besides, it has been found that there is an issue of the storage stability of modified asphalt when modifiers such as crumb rubber and mineral fillers are added into asphalt binders. Despite the better performance of modified asphalt by modification, the poor storage stability also raises concern [22,23]. There are some modification methods provided by previous literature which showed that the asphalt binders were only modified by high-speed shear and heating device in laboratory [3,18]. The poor storage stability problems might be predictable during the modification of asphalt in a mixing plant or the transportation of modified asphalt binders. Many efforts have been devoted to treating the modifiers or fillers before their utilization in the modification of asphalt binders in order to improve storage stability [14,24]. Few of these efforts have reported on either continuously modifying asphalt binders by adding crumb rubber and steel slag fillers, or preparing the compound rubber and steel slag filler modified asphalt composite (RSAC) for the purpose of better storage stability and convenient transportation of modified asphalt composite. It is important to investigate the basic and rheological behavior of RSAC when it is in the state of solid particles at room temperature.

Hence, the object of this work is to prepare the compound rubber and steel slag filler modified asphalt composite, as well as to conduct a comprehensive investigation on the rheological properties of RSAC. A base asphalt with crumb rubber was used to prepare the crumb rubber asphalt. Afterwards, the rubberized asphalt was further mixed with steel slag filler to prepare RSAC by granulation. Subsequently, the conventional properties, such as viscoelastic behavior, rheological behavior, storage stability, as well as thermogravimetric analysis were studied.

## 2. Materials

### 2.1. Base Asphalt Binder

Pen 60/80 asphalt binder was used in this research and its characteristics were concluded in Table 1 accoding to Standard Test Methods of Bitumen and Bituminous Mixtures for Highway Engineering (JTG E20-2011).

### 2.2. Crumb Rubber and Steel Slag Filler

A type of commercially available 80-mesh crumb rubber was supplied by a local company. The steel slag was collected from Wuhan Iron and Steel Corporation, Wuhan, China. Then the original slag was activated through mechanochemistry treatment for 1 h by a vertical planetary ball mill (QM-1SP2 Model, Shanghai, China). A kind of steel slag powder were used as filler in this work. The main properties of crumb rubber and steel slag filler are listed in Table 2 and Table 3 according to Asphalt Rubber for Highway Engineering (JT/T 798-2011).

## 3. Experimental Details 

### 3.1. Preparation of RSAC

Figure 1 presents a diagram of this study. The crumb rubber modified asphalt binder was prepared by mixing rubber with the base binder into a metal container and kept heated on a heating device to a temperature of 177 °C. Then, 15% (by weight of base binder) of crumb rubber and a kind of organic stabilizer was added into asphalt binder by a medium-shear radial flow impeller at a speed of 3000 rpm and a blending temperature of 177 °C for 60 min. Afterwards, the modified asphalt was kept heated at 155 °C for the subsequent experiment. Steel slag filler was blended with rubber modified asphalt. The ratio of filler to binder (F/B) was 1:1 by weight. Combined with steel slag filler, the mixtures were stirred and sheared at a speed of 750 rpm in the mixer for 60 min at a temperature of 155 °C. Then the mixtures were cooled down to room temperature. Cold and hardening mixtures developed as RSAC were crushed and ground by a ball mill and sieved to pass through a size of 9.5 cm sieve, which was finally stored in an airtight plastic bag for the subsequent experiments. Meanwhile, rubber modified asphalt (RA, 15% of rubber by weight in base binder), SBS modified asphalt (SA) and SBS modified asphalt mortar (SAM, 1:1 of F/B) as well as base asphalt (BA) were tested for comparative analysis.

### 3.2. Test Methods

A Brookfield rotational viscometer (Model DV-II+, Brookfield Engineering Inc., USA) was used to test viscosity of prepared samples at three different temperatures from 135 °C to 190 °C according to AASHTO T316. A sample of 10.5 g of RSAC and a number 27 spindle was used in this test. Rheological properties of RSAC were investigated using a dynamic shear rheometer (DSR, Anton Paar Inc., Austria) according to AASHTO T315 in terms of failure temperature, rutting factor (*G**/sin δ) and phase angle δ. A 25 mm diameter parallel plate with 1 mm gap, and 8 mm diameter parallel plate with 2 mm gap was tested at temperature range 30 °C to 80 °C, −10 °C to 30 °C for high and low temperature rheological performance respectively. In addition, thermal gravimetric analysis was performed with an integrated thermal analyzer (Netzsch, STA 449 F3, Selb, Germany). Although RSAC was prepared in order to facilitate to the transportation and storage of rubber modified asphalt, the storage stability test was necessary according to the test procedure in previous literature [25].

### 3.3. Rheological Model Analysis

The modified Christensen-Anderson-Marasteanu (CAM) model [26] was used to construct the dynamic modulus master curve of the different modified asphalt samples, as shown in Equation (1).
(1)$$G^{*}=G_{e}^{*}+\frac{G_{g}^{*}-G_{e}^{*}}{{\left[1+{\left(f_{c}/f\right)}^{k}\right]}^{m_{e}/k}}$$
where $G_{g}^{*}$ and $G_{e}^{*}$ are the glassy modulus (i.e., modulus at infinite frequency) and equilibrium modulus (i.e., modulus at zero frequency), respectively; $f_{c}$ is the crossover frequency; f is the frequency related to temperature and strain; $m_{e}$ and *k* are the non-dimensional factors. According to Equation (1), the rheological index *R* can be defined and predicted as follows.
(2)$$R=\log\frac{2^{m_{e}/k}}{1+\left(2^{\frac{m_{e}}{k}}-1\right)G_{e}^{*}/G_{g}^{*}}$$

Currently, Burgers model was widely used to simulate and characterize the viscoelasticity of asphalt binder. To better understand the creeping and recovery behavior, the Burgers model which consists of classic Maxwell and Kevin model is very suitable for describing the viscous-elastic materials such as asphalt base materials [27,28]. The equations for description of rheological behavior of RSAC in this work by Burgers model are listed as follows.
(3)$$\varepsilon \left(t\right)=\varepsilon_{el}\left(t\right)+\varepsilon_{ve}\left(t\right)+\varepsilon_{vp}\left(t\right)$$
(4)$$\varepsilon_{ve}\left(t+1\right)=\varepsilon_{el}\left(t\right)+∆\sigma /E_{0}$$
(5)$$\varepsilon_{vp}\left(t+1\right)=\varepsilon_{vp}\left(t\right)+\left(\sigma_{t}+\sigma_{t+1}\right)∆t/2\lambda_{\infty }$$
(6)$$\varepsilon_{ve}\left(t+1\right)=\varepsilon_{ve}\left(t\right)+∆\varepsilon_{ve}$$
(7)$$∆\varepsilon_{ve}=\overset{n}{\underset{i=1}{\sum }}\left[\left(\varepsilon_{ve}^{i}\left(e^{-\frac{∆t}{\tau_{i}}}-1\right)\right)+\frac{∆t}{\lambda_{i}}e^{-\frac{∆t}{2\tau_{i}}}\left(\sigma +\frac{∆\sigma }{2}\right)\right]$$
(8)$$\tau_{i}=\lambda_{i}/E_{i}$$
where, ε, $\varepsilon_{el}$, $\varepsilon_{vp}$ and $\varepsilon_{ve}$ means total, elastic, viscous-plastic and viscous-elastic strain respectively; t means time; $\lambda_{\infty }$, $E_{0}$ and σ represents the damping coefficient for the viscous part, elastic modulus, and constant stress.

## 4. Results and Discussion

### 4.1. Viscosity

The flowability of the asphalt binder can be evaluated by viscosity test which usually provides information for mixing and compaction temperatures of asphalt mixtures. In this work, the viscosity of RSAC, RA, and SAM were measured to obtain the correlation between viscosity and temperature. The results are shown in Table 4.

As can be seen from Table 4, the viscosity of all samples decreased rapidly with an increase of test temperature. The variation of its viscous-temperature curve comforted to the form of exponential function relationship model by regression analysis with coefficient ranging from 0.9687 to 0.9976. The model parameter listed in Table 4 showed that the viscosity of RSAC increased exponentially with a decrease of the temperature, and that its correlation is the best among three kinds of samples. It indicated that the temperature dependency of viscosity for all asphalt samples is significant in this study. A similar trend for viscosity can be found in previous research [29,30]. In addition, the additives such as crumb rubber and filler mixed with the asphalt binders leaded to a significant increase of viscosity and a decrease of viscosity-temperature susceptibility. Compared with RA and SAM, the higher viscosity value of RSAC was obtained because of the effect of crumb rubber and mineral filler particles in the asphalt matrix. Similar results were found in the previous research [31,32]. 

When further analyzing the viscous-temperature characteristic, the most widely used regression formula which Walther and Saal have reported, was listed as follows [33].
(9)$$\log\left(\log\eta \times 10^{3}\right)=n-m\log\left(T+273.13\right)$$
where η is viscosity; *T* is temperature; *m* and *n* mean regression coefficients; while *m* is the slope of the viscosity–temperature line, representing the temperature susceptibility, which is the viscosity temperature index. According to the viscosity test results, the regression line could be obtained as shown in Figure 2. 

The lower absolute value of *m* represents the lower viscosity–temperature susceptibility of asphalt samples. Seen from Figure 1, The absolute value of *m* of RSAC was lower than that of RA and SAM, which meant the viscosity–temperature susceptibility of RSAC was superior to RA and SAM. This was contributed to both crumb rubber and steel slag filler, which significantly influenced the temperature-sensitive properties of RSAC compared with modified asphalt binder with one additive. The results were consistent with the previous reports [6,14,33].

### 4.2. Storage Stability

The storage stability problem of modified asphalt binder is the key technical problem in the use of asphalt/modifier blends as an actual alternative to original asphalt. Additives particles dispersed in asphalt are usually accumulated and subside to the bottom of the asphalt, which may cause the modification of asphalt binder to failure. In this study, RSAC was prepared with the aim to improve storage stability and provide a convenient transportation way for modified asphalt composite. The resulting high temperature storage stability of modified asphalt samples was measured and is shown in Table 5. 

Additives such as crumb rubber and mineral fillers generally caused the reduction of storage stability of modified asphalt samples. In this study, it was noteworthy that the softening point differences (Δ*S_T_*) of all the modified asphalt samples are below 3.0 °C. It can be concluded that the steel slag filler influenced the difference of softening point and made a slight reduction in storage stability of the modified asphalt samples. RA sample shows good storage stability resulting from the degradation of rubber and its consequent reaction with asphalt component.

### 4.3. Dynamic Rheological Properties

#### 4.3.1. Master Curve

It is noted that the change of frequency of loading significantly influences the rheological properties of asphalt base materials due to dynamic traffic load. Hence, the effect of frequency on the complex module and phase angle was evaluated in this study and master curves of three asphalt samples are shown in Figure 3. Total five asphalt samples were tested to evaluate the rheological properties. SBS modified asphalt (SA) and base Pen 60/80 asphalt binder (BA) were tested as blank samples.

Seen from Figure 3, with an increase of frequency, the complex modulus of all asphalt samples increased. Besides, additives such as crumb rubber or mineral filler resulted in the complex modulus were significantly higher than that of BA and SA binders. Compared with SAM, the high temperature or low temperature of RSAC was better due to a higher complex modulus at lower or higher frequency, respectively. Meanwhile, the addition of steel slag filler in rubber modified asphalt led to a decrease in sensitivity of the frequency of asphalt materials compared with RA sample. When additives are involved, the master curve for phase angle could be more effective to evaluate the function of additives inside the modified binder. The phase angle master curve of modified asphalt samples containing polymer showed downward trend at low frequency zone compared to BA sample. However, among the RSAC, SAM, BA, and SA samples, the addition of the mineral filler in modified asphalt samples led to a decrease in the values of phase angle difference at the low frequency zone. Consequently, the phase angle downward trend disappeared and the shape of phase angle curve shown was more like that of the neat binder. Hence, it was concluded that addition of steel slag filler increased the viscosity of RSAC due having a higher phase angle, while increasing the content of elastic materials in RSAC by adding crumb rubber led to a decrease of the phase angle.

#### 4.3.2. CAM model analysis 

Based on a fitting plot by the CAM model, the parameters of model calculated according to Equations (1) and (2) are listed in Table 6. The value of *R*^2^ ranged from 0.9980 to 0.9995, which showed the master curve could be well-fitted by the CAM model. It was found that the *G_g_** of RSAC was higher than that of other samples, which means the better anti-deformation ability of RSAC at low temperature can be obtained. A higher value of *f_c_* indicated that the low temperature performance of RSAC was better than that of other samples. Shape parameters, *m_e_* and *R*, are connected with the sensibility of frequency. In this study, the lower value of *m_e_* and *R* indicated that RSAC was slightly more related to the sensibility of frequency compared with other samples. Besides, the lower value of *R* also showed that RSAC was better at a low temperature performance. Combined with Figure 3, it was concluded that the addition of crumb rubber and steel slag filler increased the complex modulus in both of low frequency and high frequency regions for modified asphalt composite when compared with BA and SA asphalt binders. However, addition of steel slag filler led to decreasing the complex modulus at low frequency, while increasing the complex modulus at high frequency when comparing RSAC with RA. The higher complex modulus of SAM could be obtained only at a high frequency region, which also indicated that RSAC was slightly influenced by the sensibility of frequency.

### 4.4. Voscous-Elastic Properties

#### 4.4.1. Creep Deformation

Three modified asphalt samples were subjected to the creep test under the two test stresses (300 Pa and 5000 Pa) at 0 °C and 60 °C, respectively. The test results are compiled in Figure 4. The total deformation, permanent deformation, and deformation recovery rate were calculated and listed in Table 7. The creep recovery rate of RSAC was slightly higher than that of RA and SAM under a 0 °C temperature, indicating that the delay elasticity of RSAC could be significantly increased at low temperature after the addition of crumb rubber and steel slag filler. Under high temperature, the recovery rate of RSAC was significantly higher than that of SAM while lower being than that of RA, which indicated that the addition of rubber led to better delay elasticity as well as total deformation and permanent deformation, representing a better anti-permanent deformation performance of rubber modified asphalt composite. As shown in Figure 4, an increase of the stress level and temperature resulted in a decrease in the deformation recovery rate of SAM and RSAC and an increase in that of RA, indicating the enhanced reducing effect towards stress sensitivity caused by the steel slag filler addition into rubber modified asphalt binder.

#### 4.4.2. Stiffness Modulus

Although the change of strain with time of asphalt samples can be observed through creep curve, as well as the viscous-elastic properties, the changing of stiffness over time is still unknown. The creep stiffness of modified asphalt samples at 0 °C and 60 °C was calculated according to Van der Pool and shown in Figure 5. The creep stiffness of RSAC and RA was significantly higher than that of SAM under high temperature, indicating the better anti-permanent deformation performance of rubber modified asphalt composite. Under low temperature, little difference between three modified asphalt samples could be observed from the curve, which meant a similar flexibility of modified asphalt samples could be obtained. In order to better understand the relationship between stiffness modulus and loading time, a linear regression according to Equation (10) was carried out, and the parameters of fitting results are listed in Table 8.
(10)$$S\left(t\right)=Bt^{-m}$$
where, *S*(*t*) is stiffness modulus, *m* is rate of variation of stiffness modulus, and *B* is a coefficient.

According to data in Table 7, the value of *R*^2^ ranged from 0.9944 to 0.9988, which showed the relationship between stiffness modulus and loading time was well-fitted by its linear regression equation, indicating the parameters *m* and *B* could be used to characterize the viscous-elastic properties of modified asphalt samples. Under a high temperature (60 °C) range, the higher value of *B* and lower value of *m* indicated that the ability of shear-creep resistance of RSAC and RA was superior to SAM. When tested under low temperature, both of B and m were lower than that of SAM, indicating that the better low temperature performance could be predictable.

#### 4.4.3. Burgers Model Analysis

It is noted that the Burgers model which was defined as a viscous-elasticity mechanics model, was widely used in analysis of viscous-elasticity materials, such as asphalt materials. Creep deformation, creep recovery, and stress relaxation of such materials can be explained by this model. In this study, fitting values of each parameter according to Equation (3) to (8) were listed in Table 8. The value of *R*^2^ varied from 0.9974 to 0.9992, which showed that the viscous-elasticity performance was well-described by a second order Burgers model. In a low temperature region, time of stress relaxation was investigated to explain the low-temperature performance of asphalt materials during creep test. The value of *λ_∞_*/*E*_0_ could be used to represent time of stress relaxation. As shown in Table 9, the value of *λ_∞_*/*E*_0_ of RSAC was lower than that of SAM and RA, indicating that the better low-temperature performance of RSAC was observed due to its smaller stress relaxation time. When a creep test was conducted under high temperature, it was found that the value of *λ_∞_* was related to permanent deformation resistance of asphalt materials. Besides, the rutting depth of asphalt mixtures not only depended on the damping coefficient in the Maxwell model, but also corresponded to the elastic modulus in the Kevin model [34]. The value of *λ_∞_* and *E*_0_ of RSAC was significantly higher than that of SAM, indicating that a better shear-creep resistance ability could be obtained. The fitting results from the second order Burgers model were consistent with the experimental results from the creep test. Additionally, it was founded that the steel slag filler showed no obviously positive effect on the performance of the modified asphalt mastic in both the creep tests and model fitting analysis when RSAC was compared with RA.

### 4.5. Thermogravimetric Analysis

Thermogravimetric analysis has been commonly used to investigate the thermal stability of asphalt binders. TG-DTG results of RSAC are depicted in Figure 6. The curve suggests that RSAC was subjected to continuous mass loss at 84.5 °C and higher temperatures. It can be seen that RSAC shows three main endothermic peaks which occurred at 84.5 °C, 460.5 °C, and 715.5 °C. These three weight loss stages were corresponded to the weight loss rate of 2.27%, 49.25%, and 7.79%, indicating loss of moisture evaporation, decomposition of asphalt binders, and decomposition of inorganic matters from mineral fillers respectively.

## 5. Conclusions

This study investigated the rheological properties of compound rubber and steel slag filler modified asphalt composite (RSAC). Conclusions are drawn as follows.

The RSAC particles were prepared to improve the storage stability and convenient transportation of modified asphalt composites. The viscosity–temperature susceptibility of RSAC was superior to that of modified asphalt binder with only one additive.

Steel slag filler led to a slight reduction in storage stability of the modified asphalt samples, while crumb rubber enhanced storage stability resulting from the degradation of rubber and its consequent reaction with asphalt component. The addition of steel slag filler in rubber modified asphalt led to a decreasing sensitivity of frequency of asphalt materials.

The creep test illustrated a better anti-permanent deformation performance of RSAC can be obtained, which means a better low-temperature performance could become predictable. The complex modulus and viscous-elasticity performance of RSCA can be well-described by the CAM and Burgers models due to their higher coefficient values.

## Figures and Tables

**Figure 1 materials-12-02588-f001:**
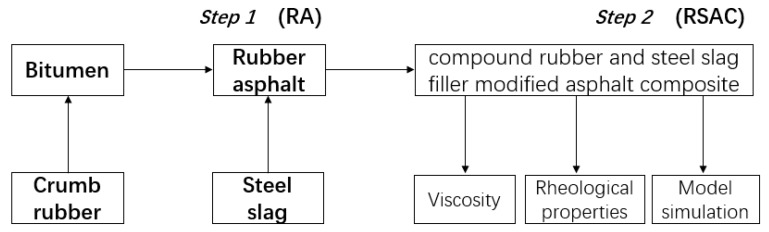
The flowchart for this study.

**Figure 2 materials-12-02588-f002:**
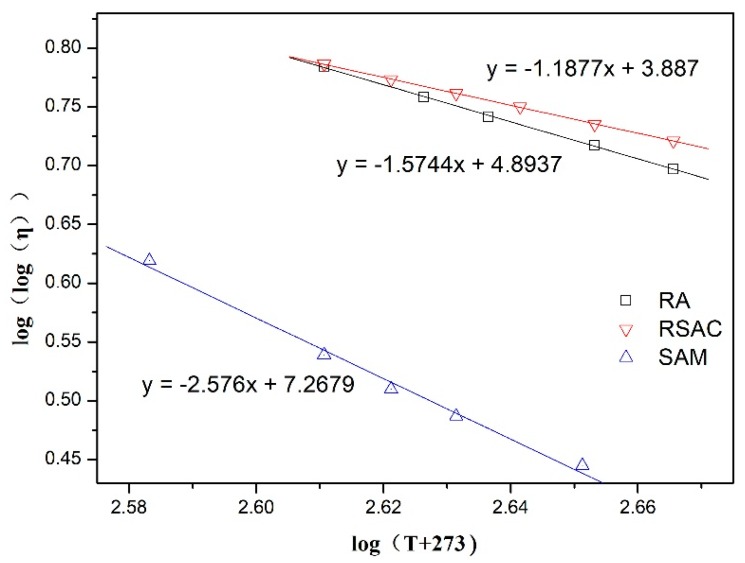
Double logarithmic type viscosity temperature curve of asphalt samples.

**Figure 3 materials-12-02588-f003:**
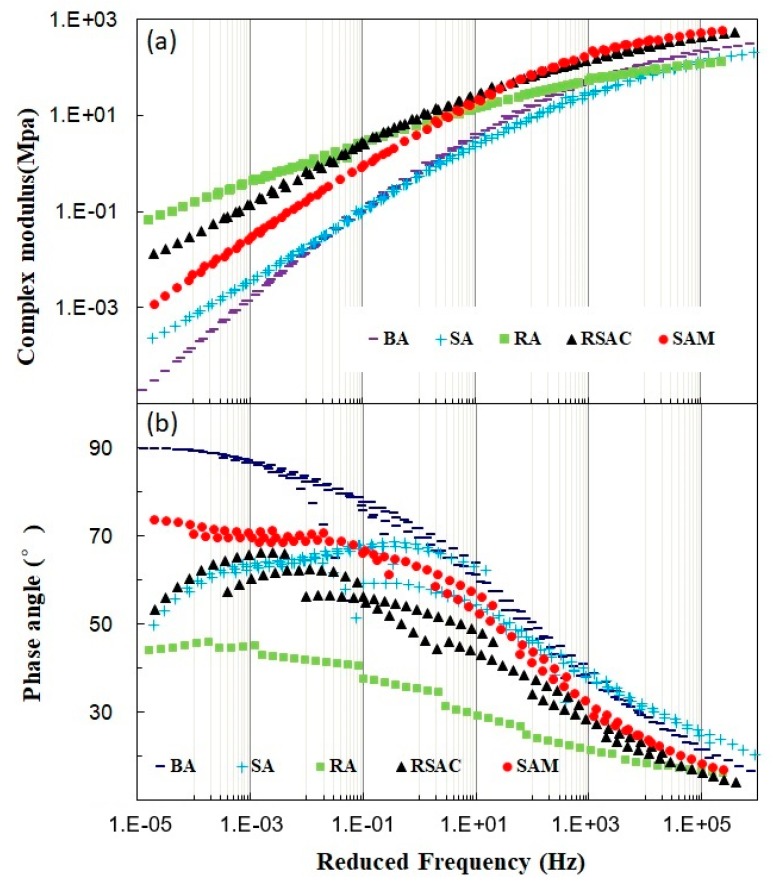
Master curves for (**a**) complex modulus and (**b**) phase angle of five asphalt samples.

**Figure 4 materials-12-02588-f004:**
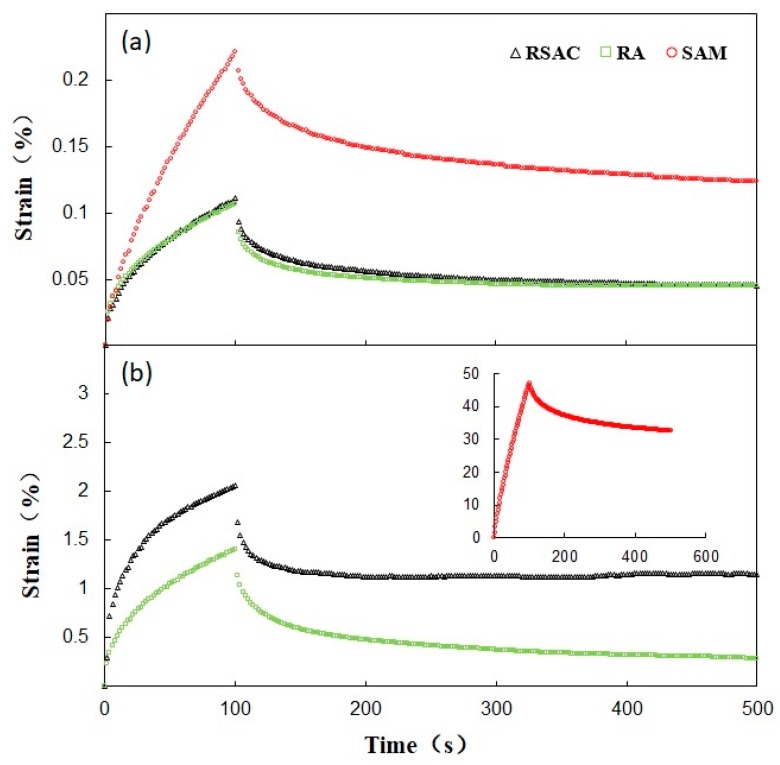
Creep deformation curves for modified asphalt samples (**a**) at 0 °C; (**b**) at 60 °C.

**Figure 5 materials-12-02588-f005:**
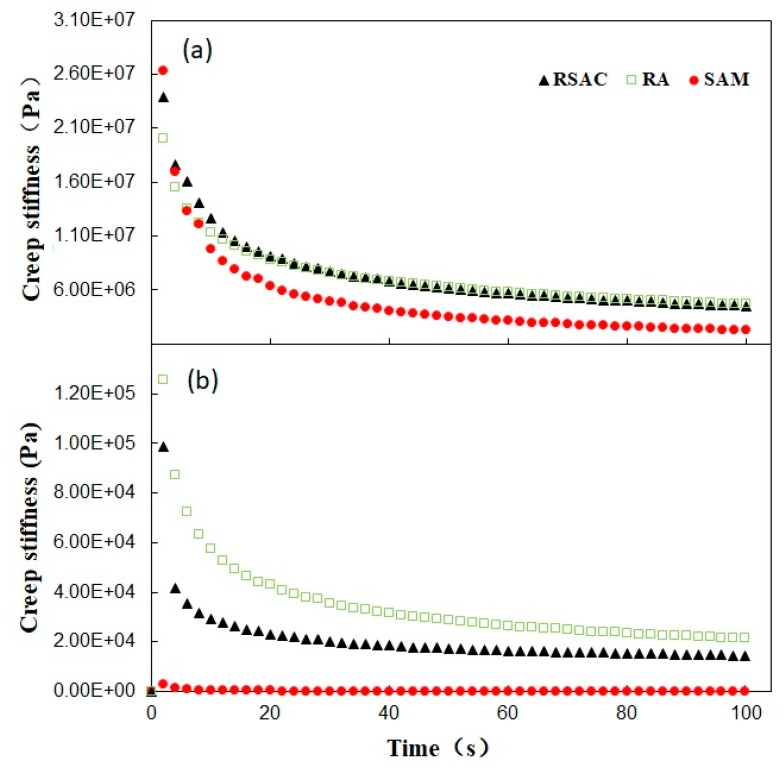
Creep stiffness curves for modified asphalt samples (**a**) at 0 °C; (**b**) at 60 °C.

**Figure 6 materials-12-02588-f006:**
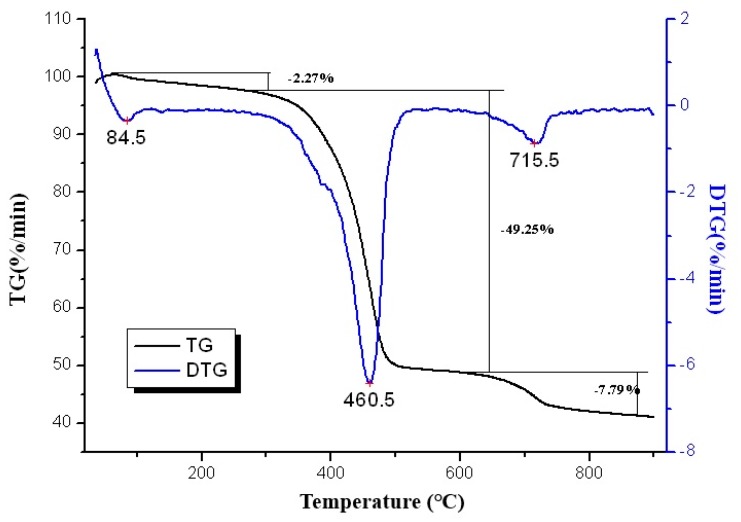
TG-DTG curves of the RSAC sample.

**Table 1 materials-12-02588-t001:** Basic properties of the asphalt binder used in this work.

Properties	Values	Specifications
Penetration [0.1 mm]	63	60–80
Penetration index	−0.7	−1.5–1.0
Softening point [°C]	47.5	≥46
Ductility, 5 cm/min, 15 °C [cm]	>160	≥100
dynamic viscosity (60 °C) [Pa·s]	179	≥160

**Table 2 materials-12-02588-t002:** Physical and chemical parameters of crumb rubber.

Properties	Values	Specifications
Density [g/cm^3^]	1.15	1.10–1.30
Moisture content [%]	0.45	<1.0
Metal content [%]	0.023	<0.05
Fiber content [%]	0.55	<1.0
Ash content [%]	4.6	≤8
Acetone extract [%]	15	≤22
Carbon black content [%]	32	≤28

**Table 3 materials-12-02588-t003:** Physical properties of steel slag.

Properties	Values	Specifications
Density [g/cm^3^]	3.47	≥2.5
Hydrophilic coefficient	0.65	<1.0
Water absorption [%]	0.65	≤1.0
Specific surface area [m^2^/g]	1.88	—

**Table 4 materials-12-02588-t004:** Viscosity of different asphalt binders and mortars at different temperatures.

Item	Viscosity (Pa·S)						Regression	
	190 °C	177 °C	160 °C	150 °C	145 °C	135 °C	Equation	*R* ^2^
RSAC	183	273	420	593	857	1310	y_1_ = 545261e(−0.0458x)	0.9976
RA	95	165	330	537	-	1220	y_2_ = 153691e(−0.0357x)	0.9954
SAM	-	0.61	-	1.17	1.72	2.89	y_3_ = 2526.4e(−0.0489x)	0.9687

**Table 5 materials-12-02588-t005:** The results of storage stability.

Item	RSAC	RA	SAM
Δ*S_T_* (°C)	2.1	1.9	2.4

**Table 6 materials-12-02588-t006:** The parameters of the CAM model.

Item	*G_g_**	*f_c_*	*m_e_*	*k*	*R*	*R* ^2^
BA	877.79	0.1970	2.0721	0.1611	3.8715	0.9980
SA	1054.52	4.9043	1.2749	0.1369	2.8029	0.9981
SAM	958.28	6.7133	1.4162	0.2287	1.8645	0.9992
RSAC	1274.73	597.0916	0.6344	0.1862	1.0258	0.9995
RA	260.44	520.5459	0.4937	0.1894	0.7846	0.9993

**Table 7 materials-12-02588-t007:** The creep deformation results of three modified asphalt samples at 0 °C and 60 °C.

Item		Total Deformation	Permanent Deformation	Recovery Rate
0 °C	SAM	0.221	0.124	43.89%
	RSAC	0.111	0.045	59.46%
	RA	0.107	0.0452	57.76%
60 °C	SAM	408	378	7.35%
	RSAC	2.06	1.15	44.17%
	RA	1.4	0.289	79.36%

**Table 8 materials-12-02588-t008:** The variation rate of creep stiffness modulus of three modified asphalt samples at 0 °C and 60 °C.

Item	60 °C			0 °C		
	*B*	*m*	*R* ^2^	*B*	*m*	*R* ^2^
SAM	6135.0	1.0095	0.9969	40,711,245.1	0.6212	0.9988
RSAC	63,069.2	0.3267	0.9944	32,702,564.7	0.4256	0.9974
RA	166,826.7	0.4522	0.9982	26,116,580.7	0.3668	0.9984

**Table 9 materials-12-02588-t009:** Regression analysis of creep for modified asphalt samples in the second order Burgers model.

Item		*E* _0_	*E* _1_	*E* _2_	*λ_∞_*	*λ* _1_	*λ* _2_	*R* ^2^
0 °C	SAM	15657893	99.68183	4538933	4.55 × 10^53^	4 × 10^8^	4.73 × 10^8^	0.9974
	RSAC	16873953	14773057	455162.1	1 × 10^48^	6.65 × 10^8^	8.87 × 10^8^	0.9983
	RA	14890529	16338206	256069.6	1× 10^48^	7 × 10^8^	1.01 × 10^9^	0.9990
60 °C	SAM	61076.82	704.8897	7398.491	7938.987	10321.62	23060923	0.9981
	RSAC	45895924	35672.25	83691.96	2631828	120250.5	2073149	0.9988
	RA	126650.1	62615.05	34509.07	11414588	1161991	5416135	0.9992

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
