# Peer review of "Performance Analysis of Compound Rubber and Steel Slag Filler Modified Asphalt Composite"

_materials, 2019, doi:10.3390/ma12162588_

Round 1

Reviewer 1 Report

The article concerns the performance analysis of compound rubber and steel  slag filler modified asphalt composite. The study presents a specific, easily identifiable advance in knowledge. It is applicable and useful to the profession. The subject matter is within the scope of the journal. The methodology is sufficiently well explained. All elements of the manuscript relate logically to the study's statement of purpose.  However, the authors must make corrections and additions before publication. Please find below my detailed comments and recommendations for corrections.

Comments to the Authors:

Abstract:

1. Please underscore the scientific value added in the abstract.

2. Add some of the most critical quantitative results to the Abstract.

3. What does RSCA mean?

Introduction:

4. Line 26: Please specify the advantages offered by asphalt mixtures.

5. As rightly emphasized in the article properties of asphalt mixtures are highly related to asphalt mortar. Therefore, introduction should be supplemented with the latest research on modifier of asphalt binders and mineral filler of asphalt mortar.

Recommended literature:

Investigating influence of mineral filler at asphalt mixture and mastic scales. Int. J. Pavement Res. Technol. 11(3) (2018), 213-224. https://doi.org/10.1016/j.ijprt.2017.10.008.

Fly ash as low cost and environmentally friendly filler and its effect on the properties of mix asphalt. Journal of cleaner production, 235 (2019) 493-502, https://doi.org/10.1016/j.jclepro.2019.06.353

Nciri, N., Shin, T., Lee, H., Cho, N., 2018. Potential of Waste Oyster Shells as a Novel Biofiller for Hot-Mix Asphalt,  Appl. Sci. 8 (3), 415. https://doi.org/10.3390/app8030415.

6. Line 66 Too many references to one sentence ([3,12,15-17,22]). Please discuss the results of the research presented in the cited papers

Materials

7. Tab 1, Tab. 2 and Tab 3 – please give the numbers of the standards according to which the tests were carried out.

Experimental Details

8. The methodology is sufficiently well explained that someone else knowledgeable about the field could repeat the study although subject is quite detailed and complicated.

Results and Discussion

9. What does SAM mean?

10. Tab 4. Some results are missing, e.g. RA in 145 - explain it

11 Was 60/80 asphalt viscosity tested before modification?

12. What does SAM mean? Why is the viscosity for these samples much lower than the others?

13. What was the repeatability of the results? How many samples were tested for Viscosity?

14. The asphalt viscosity always decreases with increasing temperature. Woszuk et al. wrote a lot about the studied viscosity of asphalt. Explaining your results, refer to the published by others.

Dynamic rheological properties

15. What does SAS (line 204) mean?

Burgers model analysis

16. Fig. 6 illegible. Please correct

Thermogravimetric analysis

17. In the analysis, please focus on the temperatures to which the bitumen is heated during the production of mix asphalt

Conclusions

The conclusions are very general. Please make sure the conclusions' section underscore the scientific value added to the paper and the applicability of the results. It is better to write the conclusions point wise.

Author Response

We have underscored the scientific value in abstract. The value in this work is to prepare a kind of composite asphalt mortar by using binders, rubber and fillers. Then the properties were investigated for evaluation of utilization of this kind of composite in production asphalt mixtures.

We have added some experimental results in abstracct.

RSAC means "compound rubber and steel slag filler modified asphalt composite" which remarked as R (RUBBER) S (STEEL SLAG ) A (asphalt) C (composite) in brief.

We have added some advantages offered by asphalt mixtrues in pavement.

We have added some the latest literatures related to modified asphalt mortar.

We cited these researches to emphersize the regular preparation method for asphalt mortar. In this work, we prepared the asphalt mortar composite in solid state first then discuss the properties, Hence, we did not illustrate the results from these results.

We have added the standard number in text.

We think that this methodology is well introduced that repeated test can be conducted.

SAM means SBS modified asphalt mortar which explained in section 3.1.

In table 4, we did not conducted the viscosity test on some specific temperature. 

base Pen 60/80 asphalt binder (BA) were tested as blank samples only in rheological experiment. We did not conduct the physical properties before its modification due to lots of similar experimental resutls published.

SAM means SBS modified asphalt mortar which explained in section 3.1. The higher viscosity of RSAC was obtained because of the effect of crumb rubber and mineral filler particles in the asphalt matrix compared to SAM. Similar results were found in the previous researches.

The avarage of three test for same samples was obtained in this work.

We have cited conclusion drawn by Woszuk to describe the changing rule of viscosity with temperature.

Here we made a mistake. There is not a asphalt binder named as SAS. It is SA binder which means SBS modified asphalt binder. We considered it as blank sample when compared with other samples.

We have deleted Fig. 6 .

In viscosity test we focused on the production temperature for modified asphlat mortar. Other tests were conducted according to specific standard methods, as a result, we have not discussed properties under same temperature. We thought this is very important suggestion that we will do as recommened in our further investigation.

We have revised the conclusion part.

Reviewer 2 Report

The paper is interesting.

The reviewer would recommend the authors to double check the paper in terms of English.

A flowchart would help at the beginning to visualize the process adopted for the research.

Author Response

Thank you for your kind advise.

We have checked and revised our English writting and expression. A  flowchart was not added in this study due to the length of this article. We will consider it as a very good suggestion and add it in our further investigation.

Thank you.

Round 2

Reviewer 1 Report

The authors have taken into account some of the comments, while some of the article still needs major revisions:

1. Too many references to one sentence (or example: line 65 [18-22], line 70 [3,14,17-19,24]). Please discuss the results of the research presented in the cited papers.

2. standards should be added in the table in an additional column

3. Tab 4 - explain why no viscosity tests were performed at certain temperatures?

4. Can you determine the level of change after the addition of additives without the results of 60/80 asphalt test results? Have you previously published the results of tests on the viscosity of 60/80 asphalt from the same production batch without additions?Asphalt from another production batch may have other properties: penetration, softening point or viscosity

5. Your answer: The higher viscosity of RSAC was obtained because of the effect of crumb rubber and mineral filler particles in the asphalt matrix compared to SAM. Similar results were found in the previous researches.

Note that these viscosities vary considerably. 273 is over 400 more than 0.61. Were all of the test conditions the same? Perhaps the spindle or spindle speed has been changed?

6. Add information about the number of samples tested in the text.

7. The asphalt viscosity always decreases with increasing temperature. Woszuk et al. wrote a lot about the studied viscosity of asphalt. Explaining your results, refer to the published by others.  The literature [13]  you quote does not apply to viscosity tests.

Recommended literature:

Influence of waste engine oil addition on the properties of zeolite-foamed asphalt, Materials, 12(14), 2265, https://doi.org/10.3390/ma12142265

Application of fly ash derived zeolites in warm-mix asphalt technology. Materials, 11, 1542, DOI: 10.3390/ma11091542.

8. In the analysis, please focus on the temperatures to which the bitumen is heated during the production of mix asphalt. TG-DTG curves are for a temperature range from 0°C to 900°C, so you can do this analysis. An analysis of the above-mentioned literature will be helpful.

9. It is better to write the conclusions point wise.

Author Response

We have deleted some references.

Standaed has been added in text by the Table. We have revised the standards used in this work. 

At some certain temperature, several viscosity test were not conducted. According to fitting results, higher co-efficient was obtained. Data from tests were suficient to be used in description of visco-temperature curve. For example, P. Li et al. (Construction and Building Materials 169 (2018) 638–647) published results on viscosity and selected some certain temperatures and data can reflect the chaning rule of viscosity. As noted, construction temperature of rubber asphalt is slightly higher than typical asphalt concrete. Hence, in viscosity test, data from high temperature was selected for RSCA and RA, while low temperatures for SAM.

In this work, we did not perform tests for determination of level of change after addition of additives. We have not compare the properties such as soft point, penetration and viscosity before and after adding additives into basic asphalt.

The reason why the viscosity varied significantly is that the dosage of crumb rubber  in this work is higher than that in previous literature. with an increase of dosage of rubber, the visocosity increased significantly according to previous research (P. Li et al. (Construction and Building Materials 169 (2018) 638–647)). Besides, the coarse particles of rubber leaded to an increase of viscosity. All of the test are conducted under same conditions. RSAC is made of rubber, filler and bitumen. Hence, the viscosity of RSAC is the highest among all samples.

Meaning of all kinds of samples are remarked in Section 3.1.

We have added these two references in text.

In TG analysis, thermogravimetric properties are illustrated according to three main peaks which showed in DTG curve. This is a very good suggestion on explaination of thermal stability at temperature which bitumen was heated. Around 177 ℃, the mass loss is not significant. The mass loss mainly derived from moisture evaporation which occurs below 200 â„ƒ. Hence, we did not discuss TG at such temperature. And we will notice that in our further investigation.

Conclusion has been rewritten point by point.

Round 3

Reviewer 1 Report

The manuscript have been improved and it can be accepted for publication

One more note remained:

Line 162 - please delete literature 13 - these are not viscosity tests

Author Response

1. Line 162 - We have deleted reference [13] which is not related to viscosity test.